# All-Cellulose Nanofiber-Based Sustainable Triboelectric Nanogenerators for Enhanced Energy Harvesting

**DOI:** 10.3390/polym16131784

**Published:** 2024-06-24

**Authors:** Mengyao Cao, Yanglei Chen, Jie Sha, Yanglei Xu, Sheng Chen, Feng Xu

**Affiliations:** 1State Key Laboratory of Efficient Production of Forest Resources, Beijing Forestry University, Beijing 100083, China; caomengyao@bjfu.edu.cn (M.C.); yanglei_chen@126.com (Y.C.); shajie123@bjfu.edu.cn (J.S.); xuyanglei@bjfu.edu.cn (Y.X.); 2Beijing Key Laboratory of Lignocellulosic Chemistry, Beijing Forestry University, Beijing 100083, China; 3Guangxi Key Laboratory of Clean Pulp & Papermaking and Pollution Control, College of Light Industry and Food Engineering, Guangxi University, Nanning 530004, China

**Keywords:** triboelectric nanogenerator, electrospinning, cellulose, nanofiber, fluorinate, energy harvesting

## Abstract

Triboelectric nanogenerators (TENGs) show promising potential in energy harvesting and sensing for various electronic devices in multiple fields. However, the majority of materials currently utilized in TENGs are unrenewable, undegradable, and necessitate complex preparation processes, resulting in restricted performance and durability for practical applications. Here, we propose a strategy that combines straightforward chemical modification and electrospinning techniques to construct all-cellulose nanofiber-based TENGs with substantial power output. By using cellulose acetate (CA) as the raw material, the prepared cellulose membranes (CMs) and fluorinated cellulose membranes (FCMs) with different functional groups and hydrophobic properties are applied as the tribopositive and tribonegative friction layers of FCM/CM-based triboelectric nanogenerators (FC-TENGs), respectively. This approach modulates the microstructure and triboelectric polarity of the friction materials in FC-TENGs, thus enhancing their triboelectric charge densities and contact areas. As a result, the assembled FC-TENGs demonstrate enhanced output performance (94 V, 8.5 µA, and 0.15 W/m^2^) and exceptional durability in 15,000 cycles. The prepared FC-TENGs with efficient energy harvesting capabilities can be implemented in practical applications to power various electronic devices. Our work strengthens the viability of cellulose-based TENGs for sustainable development and provides novel perspectives on the cost-effective and valuable utilization of cellulose in the future.

## 1. Introduction

As the energy crisis and environmental pollution become more prominent, the search for clean energy and eco-friendly functional materials that can replace fossil resources has become a global research focus [1,2]. Among the diverse energy harvesting devices, the triboelectric nanogenerator (TENG) stands out for its ability to convert mechanical energy into electrical energy through the coupling effect of contact electrification and electrostatic induction [3,4]. Since its initial discovery in 2012 [5], TENG has attracted considerable attention owing to its high power output, versatile material applicability, environmental friendliness, and cost-effective fabrication process [6,7,8]. The vertical contact-separation mode emerged as the most promising and efficient among the various operating modes of TENGs due to its outstanding output power and straightforward structure [9,10,11]. The vertical contact-separation TENGs primarily consist of three components: substrate, electrodes, and friction layers. The friction layer is the core element of TENG, which is categorized as tribopositive and tribonegative. Triboelectric material properties, including the density of triboelectric charges and relative polarity, directly influence the output performance of TENG. Currently, efforts to increase the triboelectric charge density of TENGs are focused on improving the disparity of friction material properties and enhancing the effective contact area [12,13,14,15]. Common tribopositive materials encompass polyamide (PA) [16,17], polyoxymethylene (POM) [18], polymethyl methacrylate (PMMA) [19], and metallic materials. Tribonegative materials, characterized by high triboelectric negativity and good processability, include fluorinated ethylene propylene (FEP) [20,21], polydimethylsiloxane (PDMS) [7,22], polyvinylidene fluoride (PVDF) [9,23,24,25], polytetrafluoroethylene (PTFE) [22,26,27], polyvinyl chloride (PVC) [16,28,29], and polyimide (PI) [30,31]. The TENG pair assembled using these tribopositive and tribonegative materials with different triboelectric polarity exhibit good output performance. However, these materials are mainly derived from synthetic materials, which usually suffer from some issues such as un-sustainability and comparatively inferior working performance.

Lignocellulosic biomass is the most abundant renewable natural polymer resource with the advantages of widespread availability, cost effectiveness, and biodegradability [32,33]. Its efficient functionalization and utilization in the energy field are viable strategies for addressing the aforementioned problems, thereby presenting significant potential in developing new functional materials and eco-friendly electronic devices. Cellulose is the dominant component of lignocellulose, and has become one of the preferred materials for TENGs in recent years due to its biodegradability and amenability to chemical modifiability [34,35,36,37]. Importantly, the numerous hydroxyl groups on the surface of cellulose with abundant oxygen atoms are susceptible to electron loss, which makes it suitable for use as a tribopositive material of TENGs [13,38]. However, cellulose-based functional materials have drawbacks when used as friction materials, including poor flexibility, limited performance, and the complex preparation process, usually restricting their use to tribonegative materials in TENGs. In most cellulose-based TENGs described in the reported literature, cellulose materials are only employed as tribopositive electrodes, while tribonegative electrodes predominantly consist of non-biodegradable synthetic materials [39,40]. This approach fails to fully address the non-sustainability and poor working performance of TENGs. The polarity of friction materials is determined by their physicochemical properties, which are closely related to the functional groups on their surface [41]. Cellulose-based TENG was developed through microstructural design and the chemical modification of cellulose to adjust the relative polarity and surface charge density of the friction layer, thereby enhancing output performance and sustainability [42,43]. Electrospinning has become a widely utilized method in the field of TENGs for its simplicity in producing friction layer materials with nanofibrous structures. By adjusting the spinning conditions and the properties of the polymer spinning solution, it is possible to produce nanofibrous structures with controlled morphology, high surface area, significant roughness, and good flexibility [40]. Therefore, constructing all-cellulose nanofiber-based TENGs with exceptional output performance is significantly promising by preparing tribopositive and tribonegative materials with high charge density and distinct triboelectrical properties through diverse cellulose functionalization methods.

Herein, we demonstrated the feasibility of using sustainable cellulose acetate (CA) to fabricate high-performance, all-cellulose nanofiber-based TENGs through the simple approach. This approach is based on a simple strategy combining chemical functional modification and electrospinning technology to achieve environmentally friendly and efficient energy harvesting. The CA after deacetylation and subsequent modification with trichloro (1H,1H,2H,2H-perfluorooctyl) silane produces both cellulose membrane (CM) and fluorinated cellulose membrane (FCM). The FCM/CM-based triboelectric nanogenerators (FC-TENGs) are assembled using the FCM as the tribonegative friction layer and the CM as the tribopositive friction layer, respectively. Sustainable cellulose-based materials are applied to both the tribopositive and tribonegative of the TENG. The significant difference in the friction polarity of the CM and FCM allows the FC-TENG to achieve a maximum output voltage of 94 V and a current of 8.5 µA, respectively, while maintaining good stability. With a maximum output power density of 0.15 W/m^2^, the FC-TENG is able to charge the 1 μF capacitor to 6 V in just 3.5 s. As an all-cellulose nanofiber-based sustainable power source, the FC-TENG demonstrates promise for powering small electronic devices and represents a potential avenue for further developments in low-cost and sustainable energy technologies.

## 2. Materials and Methods

### 2.1. Materials and Chemicals

The cellulose acetate (CA, acetyl content = 39.8%, Mn = 50,000), acetone, and *N*,*N*-dimethylacetamide (DMAc) were purchased from Shanghai Aladdin Biochemical Technology Co., Ltd. (Shanghai, China). Trichloro (1H,1H,2H,2H-perfluorooctyl) silane was purchased from energy chemical Co., Ltd. (Shanghai, China). Conductive copper tape and double-sided tape were purchased from USA 3M Co., Ltd. (St. Paul, MN, USA). The polytetrafluoroethylene (PTFE) film and nitrocellulose (NC) film were purchased from Jiangsu Defulong Plastic Co., Ltd. (Zhenjiang, China). Aluminum foil was purchased from the local market. 

### 2.2. Preparation of the Cellulose Acetate Membrane (CAM)

The CA was dissolved in the acetone and *N*,*N*-dimethylacetamide (DMAc) mixed solution (2:1, *v*/*v*) at room temperature (25 °C) through magnetic stirring for 2 h to obtain a mass concentration of 1.6% CA solution. The CA solution was loaded at a feed rate of 0.002 mm/s onto an electrospinning apparatus (YFSP-T, Tianjin Yunfan Technology Co., Tianjin, China) with a 21-G stainless steel needle, and operated at 25 °C, 60% humidity, and 20 kV applied voltage for 3 h. The rotating drum receiver operated at a speed of 300 rpm, maintaining a 15 cm distance between the syringe needle and receiver. The continuous fibers were deposited on aluminum foil, resulting in the formation of cellulose acetate membrane (CAM).

### 2.3. Preparation of the Cellulose Membrane (CM)

The collected CAM underwent deacetylation using the 0.1 M NaOH/ethanol solution for 8 h at room temperature. After rinsing with excess DI water to remove residual sodium hydroxide and ethanol, it resulted in the formation of cellulose membranes (CMs) after drying at room temperature.

### 2.4. Preparation of the Fluorinated Cellulose Membrane (FCM)

A fluorinated modified solution was prepared by adding trichloro (1H,1H,2H,2H-perfluorooctyl) silane (0.1, 1, 2, 3 mL) to the 100 mL toluene solution, and stirring at room temperature for 10 min. The CMs were impregnated with the fluorinated modified solution for 24 h, followed by washing with ethanol and drying to obtain fluorinated cellulose membranes (FCMs). According to the concentration of the modifier, the modified cellulose membranes were named 0.1FCM, 1FCM, 2FCM, and 3FCM, respectively.

### 2.5. Fabrication of FCM/CM-Based Triboelectric Nanogenerator (FC-TENG)

The commercial PET film (15 × 5 cm^2^) served as the flexible substrate for the material. The CM and FCM were precisely cut into self-defined sizes (4 × 4 cm^2^), serving as the tribopositive and tribonegative friction layers for TENG, respectively. The FCM/CM-based triboelectric nanogenerator (FC-TENG) was fabricated by affixing one side of the tribopositive and tribonegative friction layers with conductive copper tape on both ends of a side of the PET film using double-sided adhesive. Subsequently, the PET film was folded at both ends to ensure that the tribopositive and tribonegative friction layers were facing each other. The tribonegative friction layer of FC-TENF was substituted with FCMs (0.1FCM, 1FCM, 2FCM, and 3FCM) featuring varying degrees of fluoridation, and a series of TENGs were prepared and named 0.1FC-TENG, 1FC-TENG, 2FC-TENG, and 3FC-TENG.

### 2.6. Characterizations

The morphology and surface elements of the samples were characterized by using the scanning electron microscope (SEM, SU8010, Hitachi, Tokyo, Japan) with energy-dispersive spectrometry. The chemical structure of the samples was studied by Fourier transform infrared (FTIR) spectra mode obtained by a spectrometer (Bruker Tensor II, Bruker, MA, USA) in the ATR. X-ray photoelectron spectroscopy (XPS) data were recorded on an ESCALAB 250 photoelectron spectrometer (Thermo-Fisher Scientific, Waltham, MA, USA) with Al Kα (1486.6 eV). The water contact angle (WCA) was measured by a contact angle measuring instrument (SL200KS, KINO, New York, NY, USA). The dielectric properties were characterized using a broadband dielectric spectrometer (4294A, Agilent, Santa Clara, CA, USA)

The porosities of the membranes were calculated according to Equations (1) and (2) [7]:(1)$$Porosity=\left(1-\frac{\rho_{a}}{\rho_{s}}\right)\times \%,$$
(2)$$\rho_{a}=\frac{m}{v},$$
where *ρ_a_* is the measured density of the membranes, calculated by measuring the mass (*m*) and volume (*v*) of the membranes. The membrane was cut into circular pieces using a circular cutter with a diameter of 20 mm. The thickness of the membrane was then measured using a thickness gauge, from which the volume of the membrane was calculated. The mass of the membrane was measured using an analytical balance. *ρ_s_* is the density of the materials. The density (*ρ_s_*) of the CA, deacetylated CA, and fluorinated CA were 1.3 g/cm^3^, 1.26 g/cm^3^, and 1.37 g/cm^3^, respectively.

### 2.7. Triboelectric Generation Performance Experiments

The vertical contact-separation of the TENG is typically driven by an external mechanical force. The TENGs were periodically pressed by a shaker (Shiao, China), which was controlled by a function generator (Shiao, China) and a power amplifier (Shiao, China) to control changes in the conditions affecting the TENG’s output voltage and current. The shaker applies vibration force to the TENG through mechanical vibration, with the magnitude and frequency of the applied force controlled by the signal generator and power amplifier. The voltage, current, and charge output signals of the TENG were measured using an oscilloscope (DS1202Z-E, Rigol, Suzhou, China), an electrochemical workstation (PGSTAT302N, Metrohm, Herisau, Switzerland), and an electrometer (6514, Keithley, Cleveland, OH, USA). 

## 3. Results and Discussion

Figure 1a schematically illustrates the fabrication process of the F-membrane/C-membrane-based triboelectric nanogenerator (FC-TENG). Initially, the cellulose acetate membranes (CAMs) composed of nanofibers were prepared by electrospinning cellulose acetate (CA) solution, followed by deacetylation to obtain the cellulose membranes (CMs). Subsequently, the fluorinated cellulose membranes (FCMs) with abundant fluorine groups were produced by impregnating the CMs with trichloro (1H,1H,2H,2H-perfluorooctyl) silane, followed by rinsing and drying. The CM and FCM, possessing high specific surface areas, different hydrophobicity, and distinct triboelectric properties, functioned as the tribopositive and tribonegative friction layers, respectively. The conductive copper foils served as electrodes (Figure 1b). Appendix A shows the photograph of the prepared FC-TENG, which was expected to achieve improved energy harvesting as a result of the structure design and chemical modification.

The deacetylation of the CAM resulted in an enrichment of the CM surface with free hydroxyl groups, which is more suitable for the subsequent triboelectric generation and fluorination modification. Figure 2a,b present the photos of the CAM and CM, respectively. The CAM appears white and there is no significant color change observed in the CM after deacetylation, and the thickness and density of the CM decreased from 77 μm, 0.36 g/cm^3^ for the CAM to 67 μm, 0.17 g/cm^3^, respectively. Furthermore, the morphology and structure of the CAM and CM were observed by SEM, and the results are shown in Figure 2d,e. The CAM consisted of smooth nanofibers with a diameter of approximately 70 nm, and the deacetylation process did not significantly alter the original apparent microstructure for the CM. However, the water wettability of these two samples exhibited obvious differences. As shown in Figure 2c, the CAM presented a relatively hydrophobic feature, indicated by the contact angle of 92°. In contrast, the CM became hydrophilic with a contact angle of 18°, which was due to the formation of hydrophilic hydroxyl groups during the deacetylation process of the CAM. 

The chemical structure of the CAM and CM was investigated by Fourier transform infrared spectroscopy (FTIR). Figure 2f illustrates that the CAM exhibited the characteristic peaks at the bands of 1742, 1370, and 1228 cm^−1^, which corresponded to the C=O stretching vibration, C-CH_3_ vibration, and C-O-C vibration of the acetyl group in CA, respectively, whereas the CM displayed characteristic peaks of cellulose at 3340, 2890, and 1023 cm^−1^, attributed to the O-H stretching vibration, C-H stretching vibration, and C-O-C pyranose ring backbone vibration, respectively. The appearance of the cellulose characteristic peaks and the absence of the CA characteristic peaks indicated successful deacetylation.

The CM was modified by fluorination, where the hydroxyl groups carried on its surface were changed to fluorine-containing functional groups, and the reagent impregnation remained on the surface of the membrane. Compared with the CM and CAM, the thickness and the density of the FCM increased to about 95 μm, 0.57 g/cm^3^. The morphology of the FCMs with various modification degrees was observed by SEM and the obtained images are illustrated in Figure 3a–d. Compared with the untreated CM, the surface of the FCMs showed silane aggregate formation. The 0.1FCM featured fibers that were slightly connected, while the 1FCM displayed a distinct nodular structure enveloping the fibers. The 2FCM showed fibers wrapped around each other, maintaining the fundamental fiber structure. However, in the 3FCM, the original fiber shape was essentially lost. 

The energy-dispersive spectrometry (EDS) elucidates the grafting degree and element distribution of the FCMs. The surfaces of the FCMs exhibit a uniform distribution of F and Si elements, with distribution density gradually increasing with higher modification degrees (Figure 3e–h). Nevertheless, the distribution density of F and Si elements on the surfaces of the 2FCM and 3FCM remained relatively constant. Simultaneously, the contact angle of the CM transitions from hydrophilic to hydrophobic, while the FCM exhibits increased hydrophobicity with the augmentation of the modification degree (Figure 3i–l). Interestingly, the contact angles of the 2FCM (138°) and 3FCM (140°) are virtually identical. These observations indicate that the modification and grafting on the surface of the 2FCM are approaching saturation. 

The chemical structure of the FCMs was investigated by FTIR and X-ray photoelectron spectroscopy (XPS). As shown by the FTIR spectrum in Figure 3m, when compared with the CM, the FCM exhibited new characteristic peaks at 1142 cm^−1^ for C-CF_3_, and 1234, 1189, and 1120 cm^−1^ for C-CF_2_. Additional absorption peaks at 1064 and 1012 cm^−1^ were detected, typically associated with the Si-O-Si and Si-C vibrations of Siloxane. The XPS results further validated the successful fluorination modification of FCM. As can be seen in Figure 3n, for the CM, the characteristic peaks of C and O were observed at 286 eV and 532 eV, respectively, contrasting with the FCM’s peaks at 689 eV, 833 eV for F and 103 eV, 155 eV for Si. Moreover, newly characteristic peaks appeared at 290.8 and 293 eV for the FCM, which corresponded to -CF_2_ and -CF_3_, respectively (Figure 3o,p).

The relative polarity of the triboelectric material, together with the surface charge density and material type, significantly influences the evaluation of TENG operating characteristics such as voltage, current, and power density. The porosity of the CM was increased from 72% to 86% for the CAM by acetylation modification. The increased porosity allows the friction-generating materials to increase the surface charge when they are in contact with each other, thereby promoting an increase in electrostatic induction [44]. The dielectric constant of the FCM was increased by the fluorine modification of the CM (Appendix A). This facilitates an increase in the surface charge density, which further improves the friction generation properties [45]. However, the porosity of the F membrane decreases to 58% as it becomes denser. The fluorine groups on the FCM exhibit a remarkable ability to attract electrons, enhancing the charging performance and serving as a tribonegative to augment the contact charge density. Conversely, the surface of the CM is rich in free hydroxyl groups, which can readily release electrons and acts as the tribopositive friction layer. The easily modifiable cellulose undergoes simple and efficient chemical modification to be assembled into TENG, amplifying the electrode disparity between the materials and consequently enhancing triboelectric power generation performance (Figure 4a).

As shown in Figure 4b, the working principle of the vertical contact-separation TENGs mainly relies on the combined effects of contact electrification and electrostatic induction. In the original state, with both the tribopositive and tribonegative layers (CM and the FCM) at rest, no charges are generated, and there is no potential difference between the two electrodes. When an external force is applied, the surfaces of the tribopositive and tribonegative layers come into contact, leading to charge transfer due to the contact electrification effect. As a result, opposite electrostatic charges are generated on the two friction layers. Upon the release of the external force, the friction layers re-separate and the electrons flow from the bottom to the top electrodes to balance the electrostatic field. Therefore, the external load will experience the generated electric current. Once the friction layers are completely released, the triboelectric charges are totally neutralized and the electrical signals disappear. With the re-application of an external force, the distance between the triboelectric layers decreases gradually. This reduction in potential between the two electrodes causes the open-circuit voltage to decrease from its maximum value, and electrons flow back to the original electrode, generating a negative current. When the tribonegative layer (FCM) is completely in contact with the tribopositive layer (CM), all induced charges are neutralized and the induced voltage disappears. Furthermore, the nanofiber-based membranes with high surface area and compressible porous structures are beneficial to improve the working performance of FC-TENG, because more charges will be generated on both the contact surface and the interior pores of the friction layers.

Figure 4c,d present the output voltage and current of the FC-TENGs using the FCMs with varying degrees of fluorination as tribonegative layers. The 0.1FC-TENG exhibited the output voltage and current of 47 V and 4.5 μA, respectively. However, replacing the tribonegative layer with the 2FCM led to a remarkable increase in the output voltage and current, reaching 92 V and 8.2 μA, respectively. It is noteworthy that the 3FC-TENG showed a similar output performance. This may be attributed to the nearly saturated fluoride modification for the 2FCM, and a further increase in modifier concentration for the 3FCM did not achieve significant enhancement of the output voltage and current. According to the above results, the 2FC-TENG was selected for the subsequent experiments. Figure 4e,f show the output voltage and current of the TENGs assembled from various tribonegative materials, utilizing the CM as the tribopositive layer. The polytetrafluoroethylene (PTFE) and nitrocellulose (NC), possessing a strong capability to gain electrons, are considered as ideal choices for tribonegative materials in TENGs. The TENGs incorporating PTFE or NC both exhibited high output voltage exceeding 50 V and output current surpassing 4 μA. In comparison, the TENG employing the FCM as the tribonegative layer demonstrates significantly improved output performance, achieving an output voltage of 93 V and a current of 8.2 μA, respectively. This enhancement could be attributed to the abundance of fluorine atoms on the FCM, obtained from the fluorination modification of the CM, enhancing its gaining capability. 

The impact of external force loading and working frequency on the performance of the FC-TENGs was further investigated by measuring the output voltage and current under different forces and frequencies. As shown in Figure 4g,h, the applied external force from 5 to 20 N resulted in a substantial increase in output voltage (from 43 to 94 V) and current (from 4.1 to 8.5 μA). This obvious enhancement is primarily attributed to the positive correlation existing between the external force and the effective contact area of the triboelectric layers. With an increase in external force, the contact between the tribopositive and tribonegative friction layers became more sufficient, facilitating enhanced electrostatic charge generation and higher charge density. This elevated output voltage and current of the FC-TENG. Figure 4i,j illustrate the variation in the output voltage and current of the FC-TENG as a function of compress frequency under a constant external force of 15 N. The output voltage of the FC-TENG ranged from 92 V at 4 Hz to 94 V at 12 Hz, indicating minimal sensitivity to frequency on the output voltage. This insensitivity may be attributed to the constancy of the external force, resulting in a lack of dynamic transfer processes in the open circuit state, where voltage is solely determined by the triboelectrical charge density and the separation distance. In contrast, the output current increased significantly with frequency, rising from 5 μA at 4 Hz to 8 μA at 12 Hz. The contact deformation rate of the TENG device increases with frequency, leading to an increase in the transfer of free electrons from the external circuit and a corresponding increase in current.

For practical applications, both the output power and cycle stability play critical roles in determining the working performance of FC-TENG devices. The output voltage of the FC-TENG continuously increased and the current gradually decreased as the external load resistance increased from 10^4^ to 10^10^ Ω (Figure 5a). The output power density of the FC-TENG can be calculated by Equation (3):(3)$$P_{d}=\frac{UI}{A}\;$$
where *P_d_* is power density, *U* is output voltage, *I* is output current, and *A* is effective working area. The output power density of the FC-TENG initially ascended and then gradually decreased with the increased external load resistance. Specifically, at an external load resistance of 5 × 10^5^ Ω, the FC-TENG achieved its maximum output power density of 0.15 W/m^2^ (Figure 5b). As shown in Appendix A, the FC-TENG exhibited a relatively competitive output performance when compared to other recently reported cellulose-based TENGs. The tribopositive and tribonegative materials of the FC-TENG are based on cellulose through simple chemical modification to show good performance. Furthermore, the FC-TENG exhibited robust performance in terms of durability and stability, as demonstrated by the consistent output voltage over 15,000 consecutive contact-separation cycles (Figure 5c). There was no significant attenuation in the peak output voltage, nor was there any substantial deviation in the output signal waveform. And the morphology of the FC-TENG showed no obvious change after 15,000 cycles of operation, with the membranes still firmly attached (Appendix A, Appendix A). This excellent characteristic enabled the FC-TENG to be a promising candidate for power supply applications, specifically for powering small-scale electronic devices. Nevertheless, the periodic fluctuations inherent in the output current of the FC-TENG rendered it unsuitable for directly powering conventional electronic devices that require constant direct current (DC). To overcome this problem, the output current of the FC-TENG can be stored in the energy storage device, such as the capacitor, before utilization. The equivalent circuit diagram depicted in Figure 5d illustrates this concept, where the FC-TENG is connected to a bridge rectifier and then to the capacitor under periodic external force. The alternating current (AC) produced by the FC-TENG is rectified by the rectifier and stored in the capacitor. Figure 5e depicts the voltage variation in the capacitors with different capabilities charged by the FC-TENG. The capacitors of 1 μF, 10 μF, and 22 μF were charged rapidly to 6 V in 3.5 s, 40 s, and 110 s, respectively, under an external force of 15 N and a frequency of 10 Hz. Specifically, the 10 µF capacitor was charged to 3 V through the FC-TENG to power a commercial calculator (Figure 5f, Appendix A). However, the energy stored in the capacitor was rapidly consumed, so the repetitive charging of capacitors by the FC-TENG was required for practical utilization, as demonstrated by the change in voltage of the charged capacitor before and after powering the calculator. In addition, to enhance efficiency and flexibility in practical applications, the output power can be augmented by connecting multiple FC-TENGs in series and integrating them with high-capacity energy storage devices. This configuration enables the powering of various commercial energy devices, offering a promising avenue towards the development of sustainable energy systems in the future.

## 4. Conclusions

In summary, we have successfully fabricated all-cellulose nanofiber-based triboelectric nanogenerators (TENGs) using simple chemical modification and electrospinning techniques. The CAMs underwent deacetylation and fluorination processes to produce CMs and FCMs, which were used as triboelectric friction materials for assembling the FC-TENG. Taking advantage of the significantly different triboelectric polarity and porous nanofibers microstructure, the as-prepared FC-TENG demonstrated remarkable output performance including a voltage of 94 V, a current of 8.5 μA, and a maximum power density of 0.15 W/m^2^, thus enabling the rapid charging of capacitors and subsequent powering commercial calculators. The FC-TENG has a promising potential for future applications in energy harvesting and self-powered electronics, providing an efficient and sustainable energy solution.

## Figures and Tables

**Figure 1 polymers-16-01784-f001:**
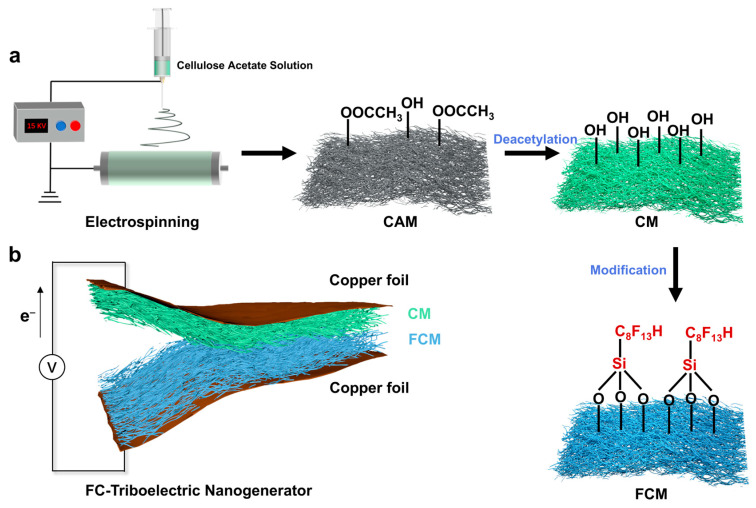
Fabrication of the FC-TENG. (**a**) The schematic illustration of the fabrication of FC-TENG. (**b**) The structure of the FC-TENG.

**Figure 2 polymers-16-01784-f002:**
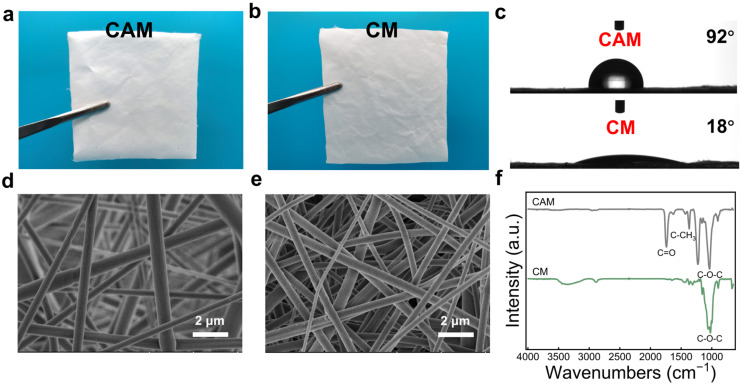
Structure and properties of the cellulose membranes. (**a**,**b**) The photographs of the CAM and CM. (**c**) The water contact angle of the CAM and CM. (**d**,**e**) The SEM images of the CAM and CM. (**f**) The FTIR spectra of the CAM and CM.

**Figure 3 polymers-16-01784-f003:**
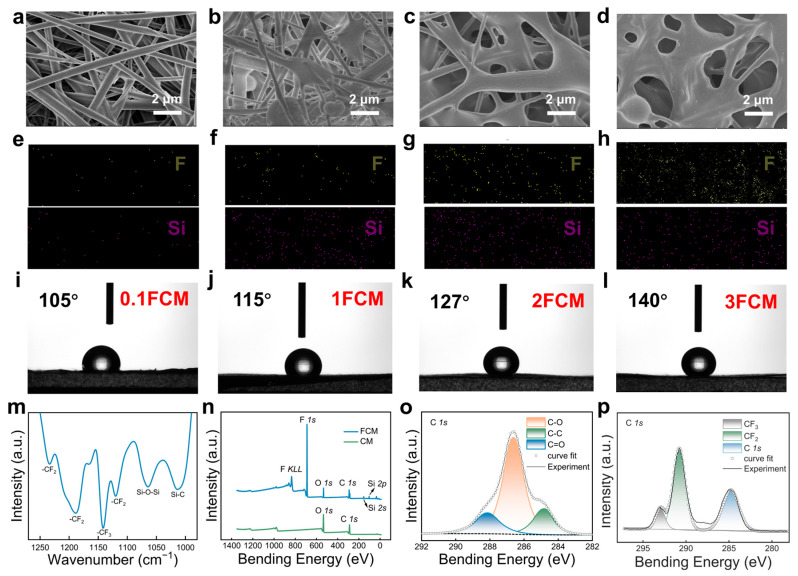
Morphology, wettability, and chemical structure of the FCMs. (**a**–**d**) The SEM images of the 0.1FCM, 1FCM, 2FCM, and 3FCM. (**e**–**h**) The EDS mapping images of the 0.1FCM, 1FCM, 2FCM, and 3FCM with F and Si. (**i**–**l**) The water contact angle of the 0.1FCM, 1FCM, 2FCM, and 3FCM. (**m**) The FTIR spectra of the FCM from 1250 to 1000 cm^−1^. (**n**) The XPS survey of the CM and FCM. (**o**) The XPS spectrum of the CM for C 1s. (**p**) The XPS spectrum of the FCM for C 1s.

**Figure 4 polymers-16-01784-f004:**
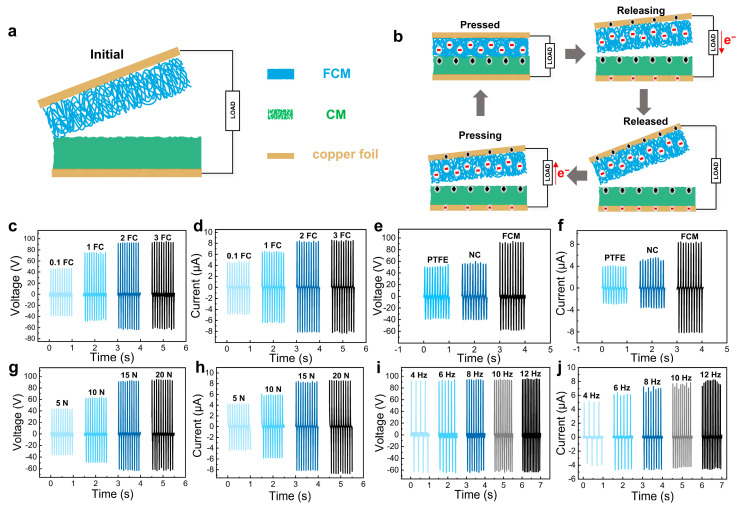
Working principle and output performance of the FC-TENG. (**a**,**b**) The schematic illustration of the structure and working mechanism of the FC-TENG. (**c**,**d**) The output voltage and current of the FC-TENG with the FCM after different degrees of fluorination. (**e**,**f**) The output voltage and current of the TENGs with the CM as tribopositive layer and different tribonegative materials. (**g**,**h**) The output voltage and current of the FC-TENG under different forces and 10 Hz. (**i**,**j**) The output voltage and current of the FC-TENG under different frequencies and 15 N.

**Figure 5 polymers-16-01784-f005:**
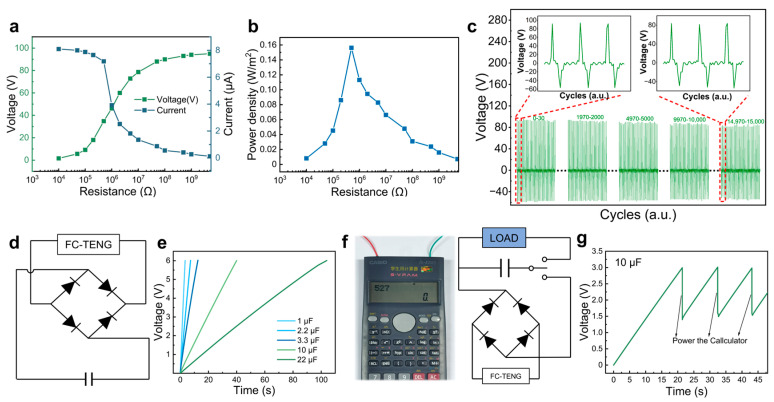
(**a**,**b**) The output voltage and power density of the FC-TENG with various external load resistances. (**c**) The output voltage of the FC-TENG under 15,000 press and release cycles. (**d**,**e**) The equivalent circuit diagram and the real-time voltage of commercial capacitors charged by the FC-TENGs. (**f**,**g**) The photograph, equivalent circuit diagram, and real-time voltage curves of the calculator powered by the commercial capacitor after FC-TENG charging.

## Data Availability

Data is available upon request from the authors due to the nature of this research.

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
