# Peer review of "All-Cellulose Nanofiber-Based Sustainable Triboelectric Nanogenerators for Enhanced Energy Harvesting"

_polymers, 2024, doi:10.3390/polym16131784_

Round 1
Reviewer 1 Report
Comments and Suggestions for Authors
Cao et al reported “All Cellulose Nanofibers-Based Sustainable Triboelectric Nan-ogenerators for Enhanced Energy Harvesting” it is an interesting work. However, a few things need to be addressed. Hence a major revision is needed.
1. In the introduction, a few latest reviews can be cited to give more details about latest developments happening in the TENG
2. A detailed literature table on cellulose-based TENG should be included to compare the performance and highlight the new features of this manuscript.
3. Photographs of the fabricated TENGs should be provided for better clarity of the device size and look. How spacing between the electrodes are maintained?
4. Usually fibers are loosely bound to the substrate in electrospinning. However, in this research fiber film itself is used as a frictional layer. Kindly provide video proof for the stability of both films.
5. Can you correlate the contact angle and TENG output?
6. Can you explain in detail step by step how current and voltage is measured as both responses look identical?
7. Why did you choose copper electrodes instead of aluminium? Copper is more prone to oxidation than aluminum.
8. Surface potential or work function or dielectric constant of the film needs to be measured to under the behavior of the TENG output.
9. Figure 5(a) should contain load current data also. When current is available power can be measured by p=(VI/A). Kindly cross your power density.
10. Author should provide the real time videos of the powering calculator
11. As the frequency is changing output voltage remains stable, however current is increasing. Do explain this part.
12. Provide the photographs of the frictional layers After 15,000 cycles stability test,
Author Response
- In the introduction, a few latest reviews can be cited to give more details about latest developments happening in the TENG
Response: Thank you for your comments, which have been very helpful in improving the article. In the introduction section, we have cited a few latest reviews and made some additions to the previous description, the changes made are marked in red. We hope that the revised manuscript meets your requirements and thank you again for your comments on this paper.
- A detailed literature table on cellulose-based TENG should be included to compare the performance and highlight the new features of this manuscript.
Response: Thanks for your insightful suggestions. We have compared the performance of the FC-TENG in our work with that of other recent reported TENGs based on cellulose in the revised manuscript and supporting information. The added text and table are as follows:
“As shown in Table S1, the FC-TENG exhibited a relatively competitive output performance when compared to other recently reported cellulose-based TENGs. The tribopositive and tribonegative materials of the FC-TENG are based on cellulose through simple chemical modification, to show good performance.” (Line355-358)
Table S1. Output performance of FC-TENG with other recent reported cellulose-based TENGs.
|
Tribopositive |
Tribonegative |
Voltage (V) |
Current (µA) |
Power density (W/m2) |
Ref. |
|
CNF Phosphorene Hybrid Paper |
Gold |
5.2 |
/ |
0.018 |
[1] |
|
Cellulose/PVDF/ BaTiO3 |
PTFE |
20.15 |
6 |
/ |
[2] |
|
Methylated superhydrophobic CNF |
FEP |
120 |
6.1 |
/ |
[3] |
|
PEO/PPG |
PCL/EC |
6.3 |
0.07205 |
2.25*10-6 |
[4] |
|
PEI/paper |
PTFE |
68.6 |
4.47 |
0.0793 |
[5] |
|
polypyrrole-coated cellulose paper |
Nitrocellulose |
60 |
/ |
0.83 |
[6] |
|
BC/BaTiO3 |
PDMS |
57.6 |
5.78 |
0.0048 |
[7] |
|
Polyamide |
PFOTES-CNF |
28.5 |
9.3 |
0.0135 |
[8] |
|
Alc-S5-CNF |
PVDF |
7.9 |
5.13 |
0.182 |
[9] |
|
CNF-PEI-Ag |
FEP |
100 |
1.1 |
0.43 |
[10] |
|
Cellulose |
Fluorinated cellulose |
94 |
8.5 |
0.15 |
This work |
* CNF = cellulose nanofiber, PVDF = polyvinylidene fluoride, PTFE = Polytetrafluoroethylene, FEP = fluorinated ethylene propylene, PEO = polyethylene oxide, PPG = poly(propylene glycol), PCL = polycaprolactone, EC = ethyl cellulose, PEI = polyethyleneimine , BC = bacterial cellulose, PDMS = polydimethylsiloxane, PFOTES = triethoxy-1H,1H,2H,2H-tridecafluoro-n-octylsilane, Alc-S5-CNF = allicin grafted CNFs.
References.
- Cui, P.; Parida, K.; Lin, M.; Xiong, J.; Cai, G.; Lee, P.S. Transparent, Flexible Cellulose Nanofibril–Phosphorene Hybrid Paper as Triboelectric Nanogenerator. Adv Materials Inter 2017, 4, 1700651, doi:10.1002/admi.201700651.
- Sun, Z.; Yang, L.; Liu, S.; Zhao, J.; Hu, Z.; Song, W. A Green Triboelectric Nano-Generator Composite of Degradable Cellulose, Piezoelectric Polymers of PVDF/PA6, and Nanoparticles of BaTiO3. Sensors 2020, 20, 506, doi:10.3390/s20020506.
- Zhang, C.; Zhang, W.; Du, G.; Fu, Q.; Mo, J.; Nie, S. Superhydrophobic Cellulosic Triboelectric Materials for Distributed Energy Harvesting. Chemical Engineering Journal 2023, 452, 139259, doi:10.1016/j.cej.2022.139259.
- Li, C.; Luo, R.; Bai, Y.; Shao, J.; Ji, J.; Wang, E.; Li, Z.; Meng, H.; Li, Z. Molecular Doped Biodegradable Triboelectric Nanogenerator with Optimal Output Performance. Adv Funct Materials 2024, 2400277, doi:10.1002/adfm.202400277.
- Wu, S.; Li, G.; Liu, W.; Yu, D.; Li, G.; Liu, X.; Song, Z.; Wang, H.; Liu, H. Fabrication of Polyethyleneimine-Paper Composites with Improved Tribopositivity for Triboelectric Nanogenerators. Nano Energy 2022, 93, 106859, doi:10.1016/j.nanoen.2021.106859.
- Shi, X.; Chen, S.; Zhang, H.; Jiang, J.; Ma, Z.; Gong, S. Portable Self-Charging Power System via Integration of a Flexible Paper-Based Triboelectric Nanogenerator and Supercapacitor. ACS Sustainable Chem. Eng. 2019, 7, 18657–18666, doi:10.1021/acssuschemeng.9b05129.
- Jakmuangpak, S.; Prada, T.; Mongkolthanaruk, W.; Harnchana, V.; Pinitsoontorn, S. Engineering Bacterial Cellulose Films by Nanocomposite Approach and Surface Modification for Biocompatible Triboelectric Nanogenerator. ACS Appl. Electron. Mater. 2020, 2, 2498–2506, doi:10.1021/acsaelm.0c00421.
- Nie, S.; Fu, Q.; Lin, X.; Zhang, C.; Lu, Y.; Wang, S. Enhanced Performance of a Cellulose Nanofibrils-Based Triboelectric Nanogenerator by Tuning the Surface Polarizability and Hydrophobicity. Chemical Engineering Journal 2021, 404, 126512, doi:10.1016/j.cej.2020.126512.
- Roy, S.; Ko, H.-U.; Maji, P.K.; Van Hai, L.; Kim, J. Large Amplification of Triboelectric Property by Allicin to Develop High Performance Cellulosic Triboelectric Nanogenerator. Chemical Engineering Journal 2020, 385, 123723, doi:10.1016/j.cej.2019.123723.
- Zhang, C.; Lin, X.; Zhang, N.; Lu, Y.; Wu, Z.; Liu, G.; Nie, S. Chemically Functionalized Cellulose Nanofibrils-Based Gear-like Triboelectric Nanogenerator for Energy Harvesting and Sensing. Nano Energy 2019, 66, 104126, doi:10.1016/j.nanoen.2019.104126.
- Photographs of the fabricated TENGs should be provided for better clarity of the device size and look. How spacing between the electrodes are maintained?
Response: Thanks for your suggestions. The photograph of FC-TENG have been provided in the Supporting Information (Figure. S1). The tribopositive and tribonegative friction layers of the FC-TENG are attached to the PET film by double-sided adhesive tape to maintain their identical spacing. The added text and figure are as follows:
“Figure S1 shows the photograph of the prepared FC-TENG, which was expected to achieve improved energy harvesting as a result of the structure design and chemical modification.” (Line191-193)
Figure S1. Photograph of the FC-TENG.
- Usually fibers are loosely bound to the substrate in electrospinning. However, in this research fiber film itself is used as a frictional layer. Kindly provide video proof for the stability of both films.
Response: Thanks for your comments. The cellulose-membranes were firmly adhered to the PET film in the preparation of the FC-TENG. During the triboelectric generation, the CM and FCM were only in vertical contact and the adhesion stability was unaffected. We have provided photographs (Figure S3) and video (Video S1) of the cellulose-membranes as the friction layers of the FC-TENG in its initial state and after 15,000 cycles of operation, and no significant changes were observed. The added text and figure are as follows:
“And the morphology of the FC-TENG showed no obvious change after 15,000 cycles of operation, with the membranes were still firmly attached (Figure S3, Video S1).” (Line362-364)
Figure S3. photographs of the FC-TENG before (a) and after (b) 15,000 recycles of operation.
- Can you correlate the contact angle and TENG output?
Response: Thanks for your comments. The friction charge density of TENG is mainly determined by the chemical potential difference on the surface of the friction material, which mainly depends on the characteristics of the functional groups of the material, thus affecting the friction generation performance [1,2]. In this work, the chemical modification method is used to modulate the chemical functional groups on the surface of the cellulose-membranes, so that the tribopositive and tribonegative materials with different functional groups. The ability to attract and release electrons is improved, thus effectively improving the friction electrical properties. The use of contact angle test can be useful to test the success of fluorination modification and the degree of modification, the results for friction power generation performance may have a certain impact, but not positive or negative correlation.
Reference:
- Nie, S.; Fu, Q.; Lin, X.; Zhang, C.; Lu, Y.; Wang, S. Enhanced Performance of a Cellulose Nanofibrils-Based Triboelectric Nanogenerator by Tuning the Surface Polarizability and Hydrophobicity. Chemical Engineering Journal 2021, 404, 126512, doi:10.1016/j.cej.2020.126512.
- Nurmakanov, Y. Structural and Chemical Modifications Towards High-Performance of Triboelectric Nanogenerators. Nanoscale Res Lett 2021, doi:10.1186/s11671-021-03578-z.
- Can you explain in detail step by step how current and voltage is measured as both responses look identical?
Response: The FC-TENG is operated by the shaker (Shiao, China), which is periodically driven at specific frequencies and external pressures to induce a periodic contact-separation motion between the tribopositive and tribonegative. The magnitude and frequency of the applied force are precisely controlled by a function generator (Shiao, China) and a power amplifier (Shiao, China). Simultaneously, the generated voltages and currents are measured and recorded by a connected oscilloscope (DS1202Z-E, Rigol, China) and electrochemical workstation (PGSTAT302N, Metrohm, Switzerland) respectively. By carefully checking the data and retesting, we have updated the graphs in Figures 4g, h, which show the output voltage and current of the FC-TENG on the effect of different applied forces. As can be seen, when the external force increases from 5 to 20 N, the output voltage and current of the FC-TENG increase accordingly. (Line172-180)
- Why did you choose copper electrodes instead of aluminium? Copper is more prone to oxidation than aluminum.
Response: Thanks for your comments. As a common electrode material in triboelectric nano-generator, copper has higher conductivity and better electronic transmission ability than aluminium, which is helpful to improve power generation efficiency and output power. However, your suggestion is indeed something we have not considered, and we will consider using other materials such as aluminium as electrode instead of copper in our subsequent work.
- Surface potential or work function or dielectric constant of the film needs to be measured to under the behavior of the TENG output.
Response: Thanks for your suggestion. The dielectric constant of the cellulose-membranes has been studied and provided in the revised manuscript and supporting information. The added text and figures are as follows:
Figure S2. Dielectric constant curves of the membranes.
“The dielectric constant of the FCM was increased by fluorine modification of the CM (Figure S3). This facilitates an increase in the surface charge density, which further improves the friction generation properties [46].” (Line264-267)
- Figure 5(a) should contain load current data also. When current is available power can be measured by p=(VI/A). Kindly cross your power density.
Response: Thanks for your comments. We have tested the output current of FC-TENG under different external load resistances, and the power density calculated by P=UI/A has been updated to in the Figure 5a, b of the revised manuscript. The modified text and figures are as follows:
Figure 5. Applications of the FC-TENG. (a), (b) Output voltage and power density of the FC-TENG with various external load resistances.
“The output voltage of the FC-TENG continuously increased and the current gradually decreased as the external load resistance increased from 104 to 1010 Ω (Figure 5a). The output power density of the FC-TENG can be calculated by equation (3):
|
, |
(3) |
where Pd is power density, U is output voltage, I is output current, and A is effective working area. The output power density of the FC-TENG initially ascended and then gradually decreased with the increased external load resistance. Specifically, at an external load resistance of 5 × 105 Ω, the FC-TENG achieved its maximum output power density of 0.15 W/m2 (Figure 5b). As shown in Table S1, the FC-TENG exhibited a relatively competitive output performance when compared to other recently reported cellulose-based TENGs. The tribopositive and tribonegative materials of the FC-TENG are based on cellulose through simple chemical modification, to show good performance.” (Line347-358)
- Author should provide the real time videos of the powering calculator.
Response: Thanks for your comments. The real time video of the powering calculator for the FC-TENG has been provided in the video S2. The added text as follows:
“Specifically, the 10 µF capacitor was charged to 3 V through the FC-TENG to power a commercial calculator (Figure 5f, Vider S2).” (Line376-378)
- As the frequency is changing output voltage remains stable, however current is increasing. Do explain this part.
Response: Different contact frequencies affect the electrostatic induction process of the TENG, thereby influencing the output electrical signal. According to the literature reports [1,2], the contact-separation frequency of the TENG generally does not affect its voltage. This is because changing the operating frequency of the TENG only alters the rate of the frictional charge transfer, while keeping the contact area and pressure constant, without affecting the total charge transferred in the external circuit. However, the shorter friction cycles will result in faster electrode transfer rates, meaning an increase in the amount of charge transferred per unit time, and consequently an increase in current. (Line330-339)
Reference:
- Zhang, Y.; Zou, J.; Wang, S.; Hu, X.; Liu, Z.; Feng, P.; Jing, X.; Liu, Y. Tailoring Nanostructured MXene to Adjust Its Dispersibility in Conductive Hydrogel for Self-Powered Sensors. Composites Part B: Engineering 2024, 272, 111191, doi:10.1016/j.compositesb.2024.111191.
- Sui, Z.; Xue, X.; Wang, Q.; Li, M.; Zou, Y.; Zhang, W.; Lu, C. Facile Fabrication of 3D Janus Foams of Electrospun Cellulose Nanofibers/rGO for High Efficiency Solar Interface Evaporation. Carbohydrate Polymers 2024, 331, 121859, doi:10.1016/j.carbpol.2024.121859.
- Provide the photographs of the frictional layers After 15,000 cycles stability test,
Response: Thanks for your comments. We have provided photographs of the cellulose-membranes as the frictional layers of the FC-TENG in its initial state and after 15,000 cycles, and no significant changes were observed. The added text and figure are as follows:
“And the morphology of the FC-TENG showed no obvious change after 15,000 cycles of operation, with the membranes were still firmly attached (Figure S3, Video S1).” (Line362-364)
Figure S3. photographs of the FC-TENG before (a) and after (b) 15,000 recycles of operation.

Reviewer 2 Report
Comments and Suggestions for Authors
The manuscript entitled "All Cellulose Nanofibers-Based Sustainable Triboelectric Nanogenerators for Enhanced Energy Harvesting". A strategy that combines straightforward chemical modification and electrospinning techniques to construct all cellulose nanofibers-based TENGs with substantial power output was proposed. In my opinion, this work is interesting and has a certain reference in the development for the application in the fields of triboelectric nanogenerators. However, there are some remarks that should be taken into consideration by the authors in order to raise this article to a good level for publication in Polymers.
The suggested modifications are listed as follows:
1. Some obvious characteristic peaks in the full spectrum of XPS are not labeled.
2. Figure 3(e)-(h) appears to be invisible.
3. The text of the illustration in Figure 5(c) is ambiguous and should be modified.
Author Response
The manuscript entitled "All Cellulose Nanofibers-Based Sustainable Triboelectric Nanogenerators for Enhanced Energy Harvesting". A strategy that combines straightforward chemical modification and electrospinning techniques to construct all cellulose nanofibers-based TENGs with substantial power output was proposed. In my opinion, this work is interesting and has a certain reference in the development for the application in the fields of triboelectric nanogenerators. However, there are some remarks that should be taken into consideration by the authors in order to raise this article to a good level for publication in Polymers.
The suggested modifications are listed as follows:
- Some obvious characteristic peaks in the full spectrum of XPS are not labeled.
Response: Thanks for your suggestion. We have labeled some distinct characteristic peaks in Figure 3(n) of the revised manuscript. The added text and modified figure are as follows:
“As can be seen in Figure 3n, for the CM, characteristic peaks of C and O were observed at 286 eV and 532 eV respectively, contrasting with the FCM’s peaks at 689 eV,833 eV for F and 103 eV,155 eV for Si.” (Line248-250)
Figure 3. (n) XPS survey of CM and FCM.
- Figure 3(e)-(h) appears to be invisible.
Response: We appreciate your valuable comments. We have improved the clarity of the Figure 3(e)-(h) to the revised manuscript. The modified figure are as follows:
Figure 3. (e-h) EDS mapping images of 0.1FCM, 1FCM, 2FCM, 3FCM with F, Si.
- The text of the illustration in Figure 5(c) is ambiguous and should be modified.
Response: Thank you for your valuable comments, which have been very helpful in improving the article. We have improved the clarity of the Figure 5(c) to the revised manuscript. The modified figure is as follows:
Figure 5. (c) Output voltage of the FC-TENG under 15,000 press and release cycles.

Reviewer 3 Report
Comments and Suggestions for Authors
Authors proposed cellulose nanofibers-based sustainable triboelectric nanogenerators for enhanced energy harvesting. Several amendments required before considering for publication:-
1-Please include clear novelty statement in last paragraph in introduction to distinguished your work with others.
2- Under preparation of the cellulose acetate membrane. please justify the temperature, humidity and hours select for the process? If from literature review, please cite accordingly.
3- Several figures such as Figure 4 (c) to (j), Figure 5(c) is not clear. Please improve the clarity.
Comments on the Quality of English LanguageAcceptable
Author Response
Authors proposed cellulose nanofibers-based sustainable triboelectric nanogenerators for enhanced energy harvesting. Several amendments required before considering for publication:
1-Please include clear novelty statement in last paragraph in introduction to distinguished your work with others.
Response: Thank you for your comments, which have been very helpful in improving the article. Our work involves the simultaneous application of sustainable cellulose-based materials to the tribopositive and tribonegative of the TENG in order to prepare high-performance, all-cellulose-based TENG. In the last paragraph of the introduction section, we have made some additions to the previous description to distinguish our work from others. The changes made in the revised manuscript are marked in red. (Line 86-88, 94-98)
2- Under preparation of the cellulose acetate membrane. please justify the temperature, humidity and hours select for the process? If from literature review, please cite accordingly.
Response: The cellulose acetate solution was kept at 25°C, 60% humidity for 3h during the electrospinning process to prepare the CAM. The process for the preparation of CAMs has been indicated in Materials and Methods in the manuscript (Line 116-119).
3- Several figures such as Figure 4 (c) to (j), Figure 5(c) is not clear. Please improve the clarity.
Response: Thank you for your valuable comments, which have been very helpful in improving the article. We have improved the clarity of the Figure 4(c)-(j) and 5(c) to the revised manuscript. The modified figures are as follows:
Figure 4. (c), (d) Output voltage and current of FC-TENG with the FCM after different degree of fluorination. (e), (f) Output voltage and current of TENGs with CM as tribopositive layer and different tribonegative materials. (g), (h) Output voltage and current of FC-TENG under different forces and 10 Hz. (i), (j) Output voltage and current of FC-TENG under different frequencies and 15 N.
Figure 5. (c) Output voltage of the FC-TENG under 15,000 press and release cycles.

Reviewer 4 Report
Comments and Suggestions for Authors
The manuscript by Mengyao Cao et al discusses the use of cellulose to create triboelectric nanogenerators. The topic discussed in the manuscript is relevant. Unfortunately, the introduction is presented in a condensed form and is intended primarily for specialists. Therefore, I recommend that the authors expand it by emphasizing the subtleties of the topic under consideration. When discussing the results, it is best to start with the characteristics of the molded membranes. I did not understand the structure of the obtained membranes F and C. Yes, the membranes were obtained, but what is their thickness, the density of the network of engagements between the fibers, what is the free volume, etc.? Without such information it is difficult to understand the results obtained. For some figures, the quality needs to be improved, for example, Figure 4.
It is advisable to expand the list of keywords.
Line 127. I recommend simplifying the designation of membranes.
The conclusions presented are concise but reflect the results obtained.
In general, my work does not cause any negative comments. The wishes presented are recommendations and not mandatory.
Author Response
- The manuscript by Mengyao Cao et al discusses the use of cellulose to create triboelectric nanogenerators. The topic discussed in the manuscript is relevant. Unfortunately, the introduction is presented in a condensed form and is intended primarily for specialists. Therefore, I recommend that the authors expand it by emphasizing the subtleties of the topic under consideration. When discussing the results, it is best to start with the characteristics of the molded membranes. I did not understand the structure of the obtained membranes F and C. Yes, the membranes were obtained, but what is their thickness, the density of the network of engagements between the fibers, what is the free volume, etc.? Without such information it is difficult to understand the results obtained. For some figures, the quality needs to be improved, for example, Figure 4.
Response: Thank you for your valuable comments. In the introduction section, we have cited the a few latest reviews and made some additions to the previous description, the changes made are marked in red.
To further characterize the structure of the obtained membranes, we have studied and provided the thickness, density, and porosity of the cellulose-membranes in the revised manuscript and supporting information. And the quality of Figure 4 has been improved. The added text (Line161-169, 197-202, 223-226, 261-268) and modified figure are as follows:
“Deacetylation of the CAM resulted in an enrichment of the CM surface with free hydroxyl groups, which is more suitable for subsequent triboelectric generation and fluorination modification. Figure 2a, b present photos of the CAM and CM, respectively. The CAM appears white and there is no significant color change observed in the CM after deacetylation, and the thickness and density of the CM decreased from 77 μm, 0.36 g/cm3 for the CAM to 67 μm, 0.17 g/cm3, respectively.” (Line 197-202)
“The CM was modified by fluorination, where the hydroxyl groups carried on its surface were changed to fluorine-containing functional groups, and the reagent im-pregnation remained on the surface of the membrane. Compared with the CM and CAM, the thickness and the density of the FCM increased to about 95 μm, 0.57 g/cm3.” (Line 223-226)
“The porosity of the CM was increased from 72% to 86% for the CAM by acetylation modification. The increased porosity allows the friction generating materials to increase the surface charge when they are in contact with each other, thereby promoting an increase in electrostatic induction [45]. The dielectric constant of the FCM was increased by fluorine modification of the CM (Figure S3). This facilitates an increase in the surface charge density, which further improves the friction generation properties [46]. However, the porosity of the F membrane decreases to 58% as it becomes denser.” (Line 261-268)
Figure 4. Working principle and output performance of the FC-TENG. (a), (b) Schematic illustration of the structure and working mechanism of FC-TENG. (c), (d) Output voltage and current of FC-TENG with the FCM after different degree of fluorination. (e), (f) Output voltage and current of TENGs with CM as tribopositive layer and different tribonegative materials. (g), (h) Output voltage and current of FC-TENG under different forces and 10 Hz. (i), (j) Output voltage and current of FC-TENG under different frequencies and 15 N.
- It is advisable to expand the list of keywords.
Thanks for your insightful suggestions. We have added keywords to the revised manuscript. The added text are as follows:
“Keywords: Triboelectric nanogenerator; Electrospinning; Cellulose; Nanofiber; Fluorinate; Energy harvesting” (Line28-29)
- Line 127. I recommend simplifying the designation of membranes.
Thanks for your insightful suggestions. we have simplified CA-membrane, C-membrane and F-membrane to CAM, CM and FCM respectively. And 0.1F, 1F, 2F and 3F membranes to 0.1FCM, 1FCM, 2FCM and 3FCM, in the revised manuscript. The changes made are marked in red.

Round 2
Reviewer 1 Report
Comments and Suggestions for Authors
The revised manuscript can be accepted in its current form.